# Serious Leisure and Passion in University Programs for Seniors

**DOI:** 10.3390/ijerph19063573

**Published:** 2022-03-17

**Authors:** Joseba Doistua, Idurre Lazcano, Aurora Madariaga

**Affiliations:** Faculty of Social and Human Sciences, University of Deusto, 48007 Bilbao, Spain; ilazkano@deusto.es (I.L.); aurora.madariaga@deusto.es (A.M.)

**Keywords:** serious leisure, passion, elderly people, learning throughout the lifespan

## Abstract

Perseverance and persistence, effort, leisure career, permanent acquisitions, specific norms, and identification with the occupation are some of the hallmarks of the serious leisure perspective. The Dualistic Model of Passion (DMP) understands passion as a strong inclination towards an activity that a person loves, considers important, and in which they invest a great amount of time and energy. This article proposes to explore these two theoretical constructs that converge in their conceptual bases, in a group of older people who regularly participate in university training programs throughout life. The sample is made up of 157 persons over 60 years old enrolled in courses for the elderly at the University of Deusto (Bilbao, Spain). The quantitative findings revealed: (1) that there are no significant differences according to sociodemographic variables, (2) that learning throughout life in the case of older people can be converted into serious leisure, and (3) that the participants in training programs for seniors develop a harmonious passion for such activity.

## 1. Introduction

### 1.1. The Serious Leisure Perspective

Leisure has been associated with predominantly positive and pleasant experiences [1]. Several investigators have noted that leisure activities appear to be especially rewarding when the person is more engaged [2,3,4]. Leisure is not only practiced for relaxation or entertainment, but it might also require determination. At times, participants have to show perseverance and strive to overcome challenges. Stebbins’ concept of serious leisure offers a relevant framework to approach these aspects of leisure [4].

Positive sociology focuses on the realm of leisure, and its conceptual roots lie for the most part in the approach known as the “serious leisure perspective” [4,5]. The serious leisure perspective (SLP) is the name of the theoretical framework that unites and synthesizes three main forms of leisure, known as serious activities, casual leisure, and project-based leisure [5]. Serious activities consist of serious leisure and devout work. The former refers to the systematic search for a main activity for an amateur, a hobbyist, or a volunteer that is highly interesting and satisfying for them and in which participants typically find a career in which they acquire and express a combination of knowledge and experience [6].

Serious leisure is distinguished from casual leisure by six characteristics that are found exclusively or are particularly elevated only in the former. These are: (1) the need to persevere in the activity, (2) the availability of a leisure career, (3) the need to strive to acquire skills and knowledge, (4) the realization of various special benefits, (5) a unique *ethos* and social world, and (6) an attractive personal and social identity. Casual leisure is immediate, inherently rewarding, and is a relatively casual activity. It is an enjoyable activity that requires little or no special training to enjoy it. It is fundamentally hedonic, and it is practiced for the significant level of pleasure found in it [7].

The adjective “serious” in this case includes traits such as sincerity and attention rather than seriousness, anguish, or anxiety [8].

Perseverance and persistence, effort, leisure career, permanent acquisitions, specific norms, and identification with the occupation are some of the hallmarks of serious participation in leisure [4,9,10,11,12]. Considering the main activities of a serious participant in leisure, university programs for seniors can be an opportunity to develop this type of leisure during adulthood. For years, multiple investigations [13,14] point out that participation in learning spaces for older adults has positive consequences such as feeling satisfaction with oneself, improving self-esteem, keeping the mind active, stimulating the intellect, achieving pleasure, enjoyment, and empowerment. In short, it improves the quality of life at all levels.

The University, through lifelong training programs, regardless of age as a discriminatory element, promotes a transformative education, not only at the educational level, but also with projections to society. These adult education programs [15] are not oriented towards achieving excellence and professional competence, but their objectives are rather focused on cultivating the mind, fostering reflection on culture [16], society and values, or simply perceiving an experience of enjoyment and satisfaction related to positive leisure experiences [17].

Furthermore, the participation of senior people in educational activities delays the onset of cognitive difficulties or neurodegenerative diseases, helps to compensate for intellectual abilities that may have begun to deteriorate [18], and even improves physical health and reduces the sense of fatigue [19]. Furthermore, participation in this type of program constitutes an opportunity to develop social skills, make friends, express oneself verbally, and become involved in a collective learning process [20,21].

### 1.2. Passion towards Leisure

Leisure is a universal phenomenon common to all people. Most people experience feelings of enthusiasm and fun, accompanied by the certainty that the activity is worth the time and effort [22]. This commitment can have a profound impact on the psychological functioning of the person [23]. For example, the study by Lapointe and Perreault [24] seeks to better understand what lies behind the motivation to participate or not in leisure activities. It is based on the self-determination theory developed by Deci and Ryan [25] with reference to relevant recent research. Two approaches based on this theory are also described: Vallerand’s hierarchical model of intrinsic and extrinsic motivation and Vallerand et al.’s dualistic model of passion.

To explain how vigorous commitment to passion influences people, Vallerand, together with other researchers, developed the Dualistic Model of Passion (DMP) and defined passion as a strong inclination towards an activity that a person loves, considers important and in which a great deal of time and energy is invested [26]. This inclination appears after a period of practice and after discovering the activities that the person experiences as satisfying or fun. Some of them will be perceived as especially important and in line with the person’s sense of identity and self-perception. These perceptions create a close relationship between the person and the activity, and, thus, the activity will turn into a passion and become an essential element of personal identity [27].

According to the DMP, there are two different types of passion: harmonious passion (HP), and obsessive passion (OP). HP is the result of the internalization of an activity -leisure in the case of this research- as part of one’s own identity, which occurs when one freely chooses to dedicate oneself to a loved activity without any pressure or contingency. Although such activity occupies an important place in the identity of the person, its presence is not overwhelming and leaves room for other vital interests in life without interfering with pursuits in other fields. Passionate involvement in an activity is totally voluntary and flexible. This type of passion fits well with other aspects and activities of a person’s life and leads to a flexible commitment. In fact, when one experiences HP, one can decide when and when not to participate in the activity. When and/or if conditions are not right, the person may suspend or even end the activity [22].

Instead, OP is the result of an internalization in one’s own identity, which occurs when the activity becomes associated with inter or intrapersonal contingencies, such as feelings of self-esteem or social acceptance [21]. OP entails an over-identification with the activity. In this state, performing the task becomes more important than other vital interests. A person with OP experiences an uncontrollable urge to participate in the activity, and even experiences conflict with other activities and interests. This conflict can also lead to negative affect and obsessive thoughts when the activity is not performed. OP also leads to rigid persistence in the activity [22].

Intrinsic motivation is the fundamental element in the learning of senior people [28,29,30], mainly derived from the satisfaction of voluntary participation [31,32] and from positive experiences [17] in a process of learning and completing those processes. This educational leisure [31] facilitates development, promoting spaces for meeting, learning, [33] and participation. Lifelong learning, therefore, contributes to giving meaning to life through achievable goals [34]. Likewise, it contributes to socialization, generating dynamics of participation, and social identification [35], with the great importance of inter or intrapersonal relationships.

Therefore, the objective of this study was to explore the results of quantitative research carried out with a group of seniors who regularly participated in university programs during their leisure time in Bilbao (Spain), focused on the perspective of serious leisure, applying the Serious Leisure Inventory and Measure (SLIM), as well as in the Dualistic Model of Passion (DMP), applying the Passion Scale. Both perspectives raise common and complementary theoretical grounds.

This study proposes that seniors who participate in university programs during their leisure time are capable of developing serious leisure activities linked to said practice (Hypothesis 1); lifelong learning as a leisure activity does not imply the development of an obsessive passion (Hypothesis 2); there are sociodemographic variables that modify the relationship of the elderly with the practice of educational leisure (Hypothesis 3).

## 2. Materials and Methods

### 2.1. Study Design and Participants

A survey was carried out in which participants were seniors enrolled in university programs for the elderly at the University of Deusto (Bilbao, Spain), who had to comply with the criterion of having completed more than one course to answer the questionnaire.

This work is part of a larger study in which the universe was made up of the total number of older people enrolled in universities in Spain. The study population was defined from data published annually by the State Association of University Programs for the elderly. The total number of people enrolled in Spanish universities during the 2019-2020 academic year was 44,994. Thus, the data show a population size of 44,994 senior students, of which 58.6% were women.

Setting an absolute error of 3%, a confidence level of 95%, and considering the hypothesis of *p* = q = 0.5, the sample size is estimated at 381 students. This article presents the data extracted from the application of the questionnaire to a sample of 162 students from the University of Deusto in Spain, the total number of students enrolled at this university who met the criteria for participation in the sample.

Data collection was carried out (in school periods) from May 2019 to February 2021. With an experimental mortality of 3.08%, the final sample comprised 157 participants with the following characteristics (Table 1):

### 2.2. Instruments

The questionnaire designed is composed of the SLIM together with the Passion Scale as well as a series of basic sociodemographic questions to know the profile of the participants and to carry out the necessary analyses to adapt the scales. The sociodemographic variables have been the following:
Sex;Year of birth;Place of residence;Amount of courses taken;Year of start for the courses.

#### 2.2.1. Serious Leisure Inventory and Measure (SLIM)

SLIM is a tool dedicated to evaluating serious leisure, a concept that explains the commitment and involvement in long-term leisure practices with which people engage in order to acquire and express the skills necessary to fully exercise those practices [8]. This tool consists of 18 dimensions derived from the six distinctive characteristics of serious leisure. Each dimension is in turn made up of three items. The items are formulated on a 9-point Likert scale in its original version and a 5-point scale in its Spanish adapted version (where 1 means “totally disagree”, and 5 means “totally agree”). Based on the 18 dimensions, the lead author of the instrument [32] derived two composite indices named “measurement” and “inventory”. The measurement index combines six of the dimensions: perseverance, effort, progress, contingencies, unique *ethos*, and identification, which refer to the different levels of seriousness that the population can show.

The inventory index combines the 12 dimensions of durable benefits. Each of the two composite indices is calculated by summing the coded responses of the items representing the relevant dimensions [36].

In this study, the Spanish version adapted by Romero, Iraurgi and Madariaga [37] has been used, who based their work on Gould’s 2008 version [9], which is composed of 54 items that respond to 18 dimensions that explain serious leisure. In this case, and given the sample age, the shorter version of 18 items (1 item per dimension) [36] has been used. The 18 items that make up this SLIM version are presented below:I overcome the difficulties in the courses I take by being persistent.I work hard to become a more competent person in the courses that I take.I feel like I have made progress in the courses I take.There are specific moments in the courses that I take that have significantly determined my involvement in them.The courses I take have enriched my life.I make full use of my talent when I take the courses.I show my abilities and skills when I take the courses.The courses I take are an expression of myself.The courses have improved the way I think about myself.The courses I take provide me with a deep sense of satisfaction.The courses I take are fun for me.I feel revitalized after classes.I have received financial reward for my experience in the courses I take.I enjoy talking to others who do the same as me.I feel important when I am part of the achievements of my group (of class).It is important to perform functions that unify my (class) group.I share many of the ideas of other people who take the same courses.Others who know me understand that these courses are a part of who I am.

The SLIM has previously offered adequate internal consistency values, and in this case, shows values ranging from 0.80 (inventory) to 0.83 (measurement).

#### 2.2.2. Passion Scale

The Passion Scale is another tool used to assess the passion people feel, referring to that strong inclination towards an activity that people like, that is considered important, and in which people invest time and energy. Therefore, for an activity to represent a passion for people, it has to hold relevance in their lives, it must be something they like, as well as something they regularly spend time on [26]. The Passion Scale was developed to measure DMP [26,38]. The scale includes two six-item subscales that assess harmonious (HP) and obsessive (OP) passion. For this study, the Spanish version of the Passion Scale [22], made up of 17 items with a Likert scale ranging from 1 (Totally disagree) to 7 (Totally agree), was used. The items that make up this scale are:The courses are in harmony with the other activities of my life.I have difficulty controlling the urge to take the courses.The new things that I discover with the courses allow me to appreciate them even more.I have almost an obsessive feeling about the courses.The courses reflect the qualities that I like about myself.The courses allow me to live a variety of experiences.The courses are the only thing that truly activates me.The courses are well integrated into my life.If I could, I would only do these courses.The courses are in harmony with other things that are part of me.The courses are so exciting that sometimes I lose control over them.I have the impression that the courses control me.I spend a lot of time practicing the courses.I like the courses.The courses are important to me.The courses are a passion for me.The courses are part of who I am.

The criterion variables have not been accounted for in this case, provided that the questions related to passion will be verified considering the sex and age of participants.

Previous studies have shown adequate internal consistency values. Results in this study oscillate between 0.76 (harmonious passion) and 0.91 (obsessive passion).

### 2.3. Procedure

Two members of the work team went to the classrooms 5 min before and/or 5 min after the classes to apply the questionnaires, explaining the purpose of the study, the importance of their participation and notifying them that they could abandon completion if they so wished. Participants had the option of taking the questionnaire during class (10 min) or taking it home and delivering it in the next class. In all cases, the questionnaire was self-administered, providing an email address for possible questions. Only on one occasion did one of the students send an email to express doubts about one of the items.

In order to be a part of the sample, it was essential that respondents stated their participation in the research was voluntary and on anonymous terms, as well as confirm they had received information on the objectives and procedures of the study and the type of participation, being able to stop their contribution at any time.

The University’s Ethics Committee, to which the researchers belong, approved this procedure in March 2019. Therefore, from the ethical point of view, the research is adequate in everything related to the protection and avoidance of risks to the participants and respect to their autonomy. Likewise, it conforms to the methodological, ethical and legal principles that this type of research should have. There were no risks of any kind for the participants and adequate measures were established that offer sufficient ethical guarantees during its development. Thus, the project takes into account the regulation on personal data protection (EU 2016/679) approved by the Commission and the EU Council in April 2016 in relation to the informed consent procedure, access to personal data, use of data for the public interest, and to the responsibilities of the researchers responsible for the project.

### 2.4. Analysis Strategy

Using the statistical program SPSS 26.0, data analysis was carried out in two phases. First, a descriptive study was carried out considering frequency and means that allowed knowing the position of the elderly regarding both scales. Then, an inferential analysis was performed through two tests: (1) an analysis of variance (ANOVA) was conducted taking into account sex and age as factor variables, as well as (2) the Brown Forsythe test [39] to correct heteroskedasticity results. In addition, the post-hoc analysis was carried out with the Scheffé test [40] to know exactly where the possible differences found in those variables of more than two categories were.

To estimate the SLIM measurement and inventory model and the harmonious and obsessive passion of the passion scale, the component items were added to each dimension dividing them by the number of total items that make up each one.

## 3. Results

### 3.1. Analysis of SLIM’s Measurement and Inventory Model

Table 2 reflects the general positioning of the people who participate in university programs for the seniors. Later on, these same results will be shown, transversed by the variables sex, age, and number of courses taken. According to participants, the most relevant dimension regards the experience of personal enrichment (4.46). Likewise, the dimensions of progress (4.13), enjoyment (4.12), satisfaction (4.09), entertainment (4.03), and group attraction (4.02) are also important (nearly reaching the maximum score of 5). Meanwhile, financing and economic achievement is the least valued dimension, showing a large difference compared with other dimensions (1.37), making it the only element with a low score worth highlighting since it is followed by the dimensions related to necessary efforts which reached an average score of 3.21.

Table 3 shows the general results regarding measurement and inventory models. The measurement index combines six of the dimensions: perseverance, effort, progress and contingencies, unique *ethos* and identification, which speak to the different levels of seriousness people can assume towards the activities they perform. The inventory index combines the 12 dimensions of durable benefits. Results show a relatively higher value of the measurement model (3.57) compared to the inventory model (2.93).

Considering the differences between gender (Table 4), the values are generally similar, with the measurement model being slightly higher in the case of men (3.59) compared to women (3.56). However, both in the models and in each of the dimensions, the differences are minimal and in no case representative (Table 5).

Regarding participants’ age, results show that the group of people over 70 years of age tends to assign higher valuations in all dimensions compared to the group of people under 65 years of age, with the only exception of the categories progress and individual expression (Table 6). So much so, that these differences are clearly reflected in the measurement and inventory models.

The post-hoc test, carried out on those variables in which significant differences between groups were found (Table 7), shows that the differences between groups in the ‘actualization’ dimension are actually found between people who are under 65 and those over 70 years old (pa-c = 0.048). In the case of the ‘Financing’ dimension, the differences are found in the three age groups analyzed (pa-b-c) = ≤ 0.001).

### 3.2. Analysis of the Dualistic Model of Passion (DMP)

Table 8 shows the means and standard deviation in the responses of the 12 items that make up the DMP, which allows observing the data independently and then analyzing the configuration of both components. Of those items that make up harmonious passion, the most highly valued were training as a reflection of the qualities they like about themselves (6.12), harmony with other activities (5.75) and with other aspects of the self (5.71), compared to variety (2.7) and discovery (2.91).

In relation to OP, the means are generally lower, where the items related to the control exercised by the courses over the person (2.2), overexcitement (2.58) and being the only thing that activates them (2.95) can be underscored for their low scores. On the contrary, two OP items reached higher scores, thus, people report having an almost obsessive feeling about the courses (5.58) and that, if they could, they would only do this type of course (5.58).

Broadly, we can state that there is a predominance of items related to harmonious passion (5.63) as evidenced compared to those related to obsessive passion (2.73). Table 9 shows that the differences between men and women are imperceptible in both components of the model, although in the case of harmonious passion the mean indicated by women (5.70) is relatively higher than in the case of men (5.49).

Results shown in Table 10 show that age does not imply significant differences in the case of obsessive passion nor in the case of harmonious passion.

## 4. Discussion

This study focuses on an aging population within the context of Western societies [41]. Given this reality, active aging is one of the central elements of the European agenda for the coming years [42]. It has been proven that greater involvement in leisure translates into higher levels of subjective well-being, a decrease in the feeling of loneliness, an increase in mood or an increase in the ability to cope with the physical and mental changes produced during aging [18,19,33,34,43].

Therefore, it is interesting to evaluate in training programs for seniors, not only the teaching and organizational quality of the curricula [29], but also the impact of the courses from other approaches focused on the personal and social development of the participants [30,32]. That is why the objectives of this paper are to evaluate the programs from the approach known as the “serious leisure perspective” or SLP [4,5], as well as from the Dualistic Model of Passion (DMP) with the profound impact on the psychological functioning of the person [23].

In this sense, the first hypothesis raised in this study is corroborated since participants in university programs for seniors show high results in the SLIM measurement model that certifies the link between said activity and serious leisure [4,5], beyond the exclusive search for benefits or enjoyment of the activity. Understanding these activities as a leisure experience in university training for seniors [31] takes on special meaning from its own design and approach.

The aspects highlighted by the practitioners of training activities for seniors in the university within the dimensions of serious leisure and that have obtained high averages such as perseverance, effort, progress, and contingencies, demonstrate those activities’ capacity to empower participants during the training process. In addition, the identification of people with a training space, together with the unique *ethos* involved in sharing many of the ideas with other people who take the same courses, are directly related to participation and empowerment, since this enhances people’s participation in the college community life. Participation is an individual act that acquires greater significance and effectiveness when it is carried out collectively, especially in groups of seniors [44].

The second hypothesis stated that lifelong learning as a leisure activity does not imply the development of an obsessive passion. The data records lower scores for obsessive passion than harmonious passion, especially in aspects related to the control exercised by the courses over people, overexcitement, and it being the only thing that activates them. However, there are two components of obsessive passion that would need to be explored further in light of high scores. These are: obsessive feelings about the courses and that, if they could, they would only do this type of course. Therefore, the initial hypothesis has been corroborated regarding the global construct, with two of its components, obsessive feelings and the exclusive choice of this type of course overloading the obsessive passion.

Finally, the third hypothesis focused on the differences in the population based on sociodemographic variables (age and gender). Specifically, it was proposed that these modify the relationship between seniors and the practice of educational leisure. In relation to serious leisure, measured through the SLIM, it has been found that participants’ age somewhat determines the degree of adherence to serious leisure, which is higher in the case of people over 70 years of age, compared to those with less than 65 years of age. So much so that these differences are clearly reflected in the measurement and inventory models. On the other hand, the differences are not significant in terms of gender. With this, the hypothesis is partially demonstrated, confirming the variations by age but not by gender. This has been verified by the post-hoc test in the dimensions of ‘Actualization’ and ‘Financing’. Continuing with this same hypothesis, the data obtained from the DMP construct leads us to reject the initial hypothesis with no significant differences found neither by gender or age groups, in any of its two dimensions, i.e., sociodemographic variables are not decisive in terms of obsessive or harmonious passion, which refer to the strong inclination towards an activity that is considered important and that demands time and energy investment.

## 5. Strengths and Limitations

This study contributes to demonstrating the relevance of evaluating whether adult training programs at university [24] meet the objectives for which they are designed. As a general rule, in these university studies, unlike others, the main objective is not exclusively educational, but rather they aim at creating a leisurely experience [5,31] and at developing human capacities to achieve human and quality community development [45]. In addition, the courses promote an experience of enjoyment and satisfaction, which results in an improvement in participants’ quality of life, without negative implications in terms of the link to the practice itself, avoiding obsessive passions. The scales presented here allow an objective evaluation of the fulfillment and optimal development of these goals.

An important contribution is considered to be the design of the used questionnaires. Completing these scales was an arduous task rooted in academia and tested in this study, and so, faced with the possibility of further analyzing these dimensions and with potential future scientific papers in the matter, we suggest they be applied, allowing not only a stable scientific base on which to lean but also the comparison of studies and future replications.

Although it is true that the results of this study are limited given the used sample and our reach to a single university with a relatively homogeneous population, the replicability of this study can allow an easy extension to other Spanish universities, which also combine different sociodemographic characteristics and that are all grouped in the State Association of University Programs for the Elderly (AEPUM for its Spanish acronym). Here lie potential future lines of work that are of high interest to the scientific community.

## Figures and Tables

**Table 1 ijerph-19-03573-t001:** Sample profile.

		Percentage
Gender	Men	39.5%
	Women	60.5%
Age	Less than 65	26.1%
	Between 65 and 70	32%
	Over 70	41.8%
Amount of courses taken	Less than 5	43%
	Between 5 and 10	38%
	Over 10	19%

**Table 2 ijerph-19-03573-t002:** Mean and standard deviation for Serious Leisure dimensions.

Dimensions	Items	M	SD
Group attraction	I enjoy talking to others who do the same as me	4.02	0.74
Entertainment	I feel revitalized after classes	4.03	0.76
Satisfaction	The courses I take give me a deep sense of satisfaction	4.09	0.72
Enjoyment	The courses I take are fun for me	4.12	0.62
Progress	I feel like I have made progress with the courses I take	4.13	0.61
Effort	I have made big efforts to become a more competent person	3.21	1.03
Financing	I have received financial reward for my experience in the courses	1.37	0.86
Group achievement	I feel important when I am part of my group’s achievements	3.40	0.94
Identity	Others who know me understand that these courses are a part of who I am	3.43	1.02
Individual expression	The courses I take are an expression of myself	3.45	0.78
Image	My way of thinking about myself has improved	3.46	0.93
Enrichment	The courses I take have enriched my life	4.46	0.58
Contingencies	There are specific moments in the courses that I take that have significantly determined my involvement	3.50	0.90
Perseverance	I overcome the difficulties in the courses I take by being persistent	3.52	0.98
Ability expression	I show my abilities and skills when I take the courses	3.52	0.77
Group maintenance	It is important to perform functions that unify my group	3.57	0.83
Unique *ethos*	I share many of the ideas of other people who take the same courses	3.65	0.82
Actualization	I make full use of my talent when I take the courses	3.86	0.78

**Table 3 ijerph-19-03573-t003:** Descriptive indicators of the measurement and inventory models.

	M	SD
Measurement model	3.57	0.64
Inventory model	2.93	0.39

**Table 4 ijerph-19-03573-t004:** Descriptive indicators for the dimensions, measurement and inventory models by gender.

	Men (*n* = 62)		Women (*n* = 95)	
	M	SD	M	SD
Perseverence	3.50	1.02	3.53	0.97
Effort	3.27	1.04	3.18	1.04
Progress	4.16	0.61	4.12	0.62
Contingencies	3.63	0.91	3.41	0.91
Enrichment	4.48	0.54	4.45	0.62
Actualization	3.90	0.62	3.82	0.88
Ability expression	3.56	0.64	3.49	0.85
Individual expression	3.45	0.72	3.44	0.83
Image	3.48	0.92	3.42	0.94
Satisfaction	4.11	0.70	4.07	0.75
Enjoyment	4.11	0.55	4.12	0.67
Entertainment	3.98	0.74	4.03	0.78
Financing	1.44	0.86	1.35	0.88
Group attraction	3.95	0.71	4.06	0.76
Group achievement	3.31	0.82	3.45	1.02
Group maintenance	3.52	0.92	3.61	0.78
Unique *ethos*	3.63	0.79	3.65	0.84
Identity	3.35	0.94	3.45	1.08
Measurement	3.59	0.60	3.56	0.67
Inventory	2.92	0.33	2.93	0.43

**Table 5 ijerph-19-03573-t005:** Summary of the ANOVA of one factor: Serious Leisure Inventory and Measure (SLIM) considering gender.

	*F*	*df*	*p*
Perseverence	0.027	1.155	0.871
Effort	0.313	1.155	0.576
Progress	0.207	1.155	0.650
Contingencies	2.178	1.155	0.142
Enrichment	0.107	1.155	0.744
Actualization	0.412	1.155	0.522
Ability expression	0.304	1.155	0.582
Individual expression	0.005	1.155	0.941
Image	0.170	1.155	0.680
Satisfaction	0.108	1.155	0.743
Enjoyment	0.001	1.155	0.977
Entertainment	0.147	1.155	0.702
Financing	0.38	1.155	0.538
Group attraction	0.856	1.155	0.356
Group achievement	0.896	1.155	0.345
Group maintenance	0.479	1.155	0.490
Unique *ethos*	0.031	1.155	0.86
Identity	0.340	1.155	0.561
Medida	0.111	1.155	0.739
Inventario	0.016	1.155	0.899

**Table 6 ijerph-19-03573-t006:** Descriptive indicators for the dimensions, measurement and inventory models by age.

	Less Than 65 *(n* = 40) (a)		65 to 70 (*n* = 49) (b)		Over 70 (*n* = 64) (c)	
	M	SD	M	SD	M	SD
Perseverence	3.43	1.05	3.31	0.98	3.69	0.92
Effort	3	1.06	3.1	1.08	3.38	0.96
Progress	4.23	0.42	4.08	0.78	4.08	0.54
Contingencies	3.38	0.95	3.51	0.98	3.53	0.83
Enrichment	4.4	0.49	4.39	0.67	4.55	0.56
Actualization	3.58	0.87	3.9	0.65	3.97	0.82
Ability expression	3.43	0.81	3.47	0.79	3.56	0.75
Individual expression	3.53	0.75	3.47	0.77	3.38	0.83
Image	3.28	1.06	3.55	0.79	3.47	0.96
Satisfaction	4	0.75	4.18	0.75	4.05	0.70
Enjoyment	4.15	0.42	4.1	0.71	4.11	0.64
Entertainment	4	0.81	3.94	0.77	4.08	0.74
Financing	1.1	0.30	1.14	0.46	1.72	1.16
Group attraction	3.98	0.69	3.9	0.87	4.09	0.66
Group achievement	3.20	1.09	3.41	0.89	3.52	0.87
Group maintenance	3.53	0.84	3.45	0.84	3.69	0.81
Unique *ethos*	3.45	0.74	3.67	0.80	3.75	0.85
Identity	3.20	1.20	3.49	0.89	3.55	0.96
Measurement	3.44	0.66	3.52	0.72	3.66	0.56
Inventory	2.84	0.34	2.87	0.40	3.01	0.41

**Table 7 ijerph-19-03573-t007:** Summary of one-factor ANOVA: Serious Leisure Inventory and Measure (SLIM) considering age.

	*F*	*df*	*p*
Perseverence	2.249	2.150	0.109
Effort	1.887	2.150	0.155
Progress	0.848	2.150	0.430
Contingencies	0.389	2.150	0.678
Enrichment	1.294	2.150	0.277
Actualization	3.249	2.125	0.042
Ability expression	0.425	2.150	0.655
Individual expression	0.482	2.150	0.619
Image	0.990	2.150	0.374
Satisfaction	0.797	2.150	0.453
Enjoyment	0.075	2.150	0.927
Entertainment	0.46	2.150	0.632
Financing	13.306	2.959	≤0.001
Group attraction	0.993	2.150	0.373
Group achievement	1.395	2.150	0.251
Group maintenance	1.215	2.150	0.300
Unique *ethos*	1.722	2.150	0.182
Identity	1.552	2.150	0.215
Measurement	1.486	2.150	0.230
Inventory	2.538	2.150	0.082

**Table 8 ijerph-19-03573-t008:** Descriptive indicators of the DMP items and components.

	Items	M	SD
Harmonious Passion	1. The courses are in harmony with the other activities in my life	5.75	1.12
	3. The new things I discover in the courses allow me to appreciate them even more	2.91	1.67
	5. The courses reflect the qualities that I like about myself	6.12	0.84
	6. The courses allow me to live a variety of experiences	2.70	1.63
	8. The courses are well integrated into my life	4.94	1.18
	10. The courses are in harmony with other things that are part of me	5.71	1.07
Obsessive Passion	2. I have difficulty controlling the urge to take courses	3.07	1.56
	4. I have an almost obsessive feeling about the courses	5.58	1.04
	7. The courses are the only thing that truly activates me	2.95	1.54
	9. If I could I would only do these courses	5.60	1.11
	11. The courses are so exciting that sometimes I lose control over them	2.58	1.51
	12. I have the impression that the courses control me	2.20	1.39
Harmonious Passion		5.63	0.71
Obsessive Passion		2.73	1.30

**Table 9 ijerph-19-03573-t009:** Summary of the ANOVA of one factor: DMP considering gender.

	Men (*n* = 62)		Women (*n* = 94)		*F*	*df*	*p*
	M	SD	M	SD			
Harmonious Passion	5.49	0.65	5.70	0.74	3.14	1.153	0.078
Obsessive Passion	2.87	1.22	2.64	1.31	1.21	1.154	0.272

**Table 10 ijerph-19-03573-t010:** Summary of the ANOVA of one factor: DMP considering age.

	Less than 65 (*n* = 40) (a)		65 to 70 (*n* = 49) (b)		Over 70 (*n* = 63) (c)		*F*	*df*	*p*
	M	SD	M	SD	M	SD			
Armonious Passion	5.54	0.74	5.57	0.71	5.74	0.70	1.21	2.148	0.301
Obsessive Passion	2.40	1.22	2.68	1.13	2.94	1.40	2.21	2.149	0.114

## Data Availability

This research was approved by the Ethics Committee of the University of Deusto (Spain) in March 2019 (code ETK-26/18-19).

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
