# Peer review of "Serious Leisure and Passion in University Programs for Seniors"

_ijerph, 2022, doi:10.3390/ijerph19063573_

Round 1

Reviewer 1 Report

First of all, this manuscript deals with a topic of interest in the field of “Leisure, Wellbeing and Health”.

The theoretical and conceptual frameworks on which the manuscript is based are elaborated in a satisfactory manner. However, it is necessary to justify the relationship among lifelong learning, university programs for senior and the inclination towards activities that people love. The authors summarize and connect previous findings but do not delve into this relationship.

The methodology is adequate and it is elaborated in a satisfactory manner.

The Discussion is well written and clear. However, regarding the second and third hypotheses, the authors merely (re-)state their outcomes without any reference to the work of others. It is important to enrich this part with other studies. 

Author Response

Sincerely, Thank you very much for all your comments and evaluation work. We have tried to improve the article by responding to your all indications.

Reviewer 2 Report

The study presents an interesting idea regarding the integration of two approaches in the investigation of the phenomenon of leisure: the Serious Leisure Perspective (SLP) and the Dualistic Model of Passion (DMP) – the second of them not originally formulated for the explanation of leisure.

However, the proposal must be deepened both in the theoretical and methodological aspects, in order to be able to answer questions such as the following:

  • What can explain the DMP that does not contemplate the SLP? What precedents are there in this regard and how do they dialogue with the proposed research?
  • How is it justified that, if the validated SLIM scale is the long version, the short version has been chosen (for which there seems to be no precedent in the Spanish language)?
  • Why have the variables of gender and age been chosen to analyze the different values ​​of the scales? What precedents are there to justify the comparison? It is striking that, being a study that proposes the complementarity of both theories, it does not offer at least one table of correlations between both instruments.

These observations seek to stimulate the authors to strengthen the main argument of the article, so that it responds to the topic of the special issue, relative to "Leisure and Time Management in Fostering Wellbeing and Health: Current Issues and New Trends". It is advisable to place greater emphasis on what the article contributes to the proposed topic.

In sum, once issues such as those mentioned have been resolved, the article may have the required quality conditions of the IJERPH.

Author Response

Sincerely, Thank you very much for all your comments and evaluation work. An attempt has been made to improve the article by responding to the indications.

The short version of the SLIM has been selected because it is validated in this English version. In addition, the items in Spanish are the same as the long version.

The sociodemographic variables have been chosen because sociodemographic data enrich and clarify any approach that is made, and this study aims to be an approach to this complementarity that other authors have already done, theoretically and in the same group.

In the conclusions we have tried to explain the dialogue between both theories, as well as the limitations present in this study, which has sought to advance knowledge knowing that there are still limitations (sample size) as well as future lines (analysis of correlations between instruments, as indicated in the review).

Reviewer 3 Report

Very interesting topic to larger audience. Especially considering ageing population in EU and western countries in general.

Here some suggestions for improvement and corrections:

Consider the title:  Serious Leisure and Passion of Seniors in University Programs

In 2. Materials and Methods:

 2.1. Study Design y Participants correct the   y ->  and

…the final sample size was 157 questionnaires distributed as follows…  Sample size comprise participants (who filled questionnaires) ; questionnaires was not sample size. Rephrase this sentence something like: the final sample size was  N=157 , or …sample size comprise 157 participants with following characteristics, or distribution…

2.4. Analysis Strategy    change to:  Statistical analysis

In table 6 and 8  equalise the number of decimal places in M values. If M=2.7  than write 2.70 because other values in table are expressed in two decimals, etc.

In  4. Discussion

This study focuses on the general aging of the population within the…  Suggestion to rewrite in this way:  This study focuses on aging population within the… 

Author Response

(The authors gave the same response as above.)

Reviewer 4 Report

The manuscript entitled "Serious Leisure and Passion in University Programs for Seniors" sets out the study on serious Leisure and passion for seniors. Although it is well structured and coherent, there are a number of elements that need to be revised:

  • The sample is made up of 157 participants. Is the sample enough to propose the conclusions in the manuscript?
  • How representative is the sample used?
  • Please explain the reasons some numbers listed after keywords. If they are typos. Please delete the numbers after keywords.
  • The objective should be better defined and stated just before starting to describe the material and method section.
  • Please check the title of section 2.1. What does y mean?
  • Please check the title of “strengths” and limitation
  • Does criterion (in 2.2.2) mean dimension (in 2.2.1)? If they are the same, please use the same word to keep consistency
  • The discussion must be better detailed

Author Response

Sincerely, Thank you very much for all your comments and evaluation work. In the case of the objective, we have focused more on the hypothesis statement, so it is the hypothesis that ends section 1.

Although it is true that the results of this study are limited given the used sample and our reach to a single university with a relatively homogeneous population, the replicability of this study can allow an easy extension to other Spanish universities.
